# Direct targets of *MEF2C* are enriched for genes associated with schizophrenia and cognitive function and are involved in neuron development and mitochondrial function

**Deema Ali**[1,2], **Aodán Laighneach**[1,2], **Emma Corley**[1,3], **Saahithh Redddi Patlola**[1,4], **Rebecca Mahoney**[1,2], **Laurena Holleran**[1,3], **Declan P. McKernan**[4], **John P. Kelly**[4], **Aiden P. Corvin**[5], **Brian Hallahan**[1,6], **Colm McDonald**[1,6], **Gary Donohoe**[1,3], **Derek W. Morris**[1,2]*

1 Centre for Neuroimaging, Cognition and Genomics (NICOG), University of Galway, Ireland, 2 School of Biological and Chemical Sciences, University of Galway, Ireland, 3 School of Psychology, University of Galway, Ireland, 4 Discipline of Pharmacology & Therapeutics, School of Medicine, University of Galway, Ireland, 5 Neuropsychiatric Genetics Research Group, Department of Psychiatry, Trinity College Dublin, Ireland, 6 Discipline of Psychiatry, School of Medicine, University of Galway, Ireland

* derek.morris@universityofgalway.ie

**Data Availability Statement:** The primary data underlying the results presented in this study are available from a study that have already been

## Abstract

*Myocyte Enhancer Factor 2C* (*MEF2C*) is a transcription factor that plays a crucial role in neurogenesis and synapse development. Genetic studies have identified *MEF2C* as a gene that influences cognition and risk for neuropsychiatric disorders, including autism spectrum disorder (ASD) and schizophrenia (SCZ). Here, we investigated the involvement of *MEF2C* in these phenotypes using human-derived neural stem cells (NSCs) and glutamatergic induced neurons (iNs), which represented early and late neurodevelopmental stages. For these cellular models, *MEF2C* function had previously been disrupted, either by direct or indirect mutation, and gene expression assayed using RNA-seq. We integrated these RNA-seq data with *MEF2C* ChIP-seq data to identify dysregulated direct target genes of *MEF2C* in the NSCs and iNs models. Several *MEF2C* direct target gene-sets were enriched for SNP-based heritability for intelligence, educational attainment and SCZ, as well as being enriched for genes containing rare *de novo* mutations reported in ASD and/or developmental disorders. These gene-sets are enriched in both excitatory and inhibitory neurons in the prenatal and adult brain and are involved in a wide range of biological processes including neuron generation, differentiation and development, as well as mitochondrial function and energy production. We observed a trans expression quantitative trait locus (eQTL) effect of a single SNP at *MEF2C* (rs6893807, which is associated with IQ) on the expression of a target gene, *BNIP3L*. *BNIP3L* is a prioritized risk gene from the largest genome-wide association study of SCZ and has a function in mitophagy in mitochondria. Overall, our analysis reveals that either direct or indirect disruption of *MEF2C* dysregulates sets of genes that contain multiple alleles associated with SCZ risk and cognitive function and implicates neuron development and mitochondrial function in the etiology of these phenotypes.

published (Mohajeri et al, 2022) at DOI: 10.1016/j.ajhg.2022.09.015, which has been referenced in the manuscript. The data link is: https://www.ncbi.nlm.nih.gov/geo/query/acc.cgi?acc=GSE204778.

**Funding:** This work was funded by grants from the University of Galway, Ireland (https://www.universityofgalway.ie/hardiman-scholarships/; Hardiman Research Scholarship #128936 to DA), the European Research Council (https://erc.europa.eu/; ERC-2015-STG-677467 to GD) and Science Foundation Ireland (https://www.sfi.ie/; SFI 16/ERCS/3787 to GD). The funders did not play any role in the study design, data collection and analysis, decision to publish, or preparation of the manuscript.

**Competing interests:** The authors have declared that no competing interests exist.

## Author summary

Schizophrenia is a complex disorder caused by many genes. Current drugs for schizophrenia are only partially effective and do not treat cognitive deficits, which are key factors for explaining disability, leading to unemployment, homelessness and social isolation. Large-scale genetic studies of schizophrenia and cognitive function have been effective at identifying individual SNPs and genes that contribute to these phenotypes but have struggled to immediately uncover the bigger picture of the underlying biology of the disorder. Here we take an individual gene associated with schizophrenia and cognitive function called *MEF2C*, which on its own is a very important regulator of brain development. We use functional genomics data from studies where *MEF2C* has been mutated to identify sets of other genes that are influenced by *MEF2C* in developing and mature neurons. We show that several of these gene-sets are enriched for common variants associated with schizophrenia and cognitive function, and for rare variants that increase risk of various neurodevelopmental disorders. These gene-sets are involved in neuron development and mitochondrial function, providing evidence that these biological processes may be important in the context of the molecular mechanisms that underpin schizophrenia and cognitive function.

## Introduction

*MEF2C*, a transcription factor within the myocyte enhancer factor-2 (MEF2) family, is involved in essential neurodevelopmental processes [1]. *MEF2C* is expressed during the initial stages of embryonic brain development and remains expressed at elevated levels in adult brains, including in the striatum, hippocampus, and cortex, indicating an involvement in both embryonic and adult brain activity [1,2]. *MEF2C* plays a critical role in neurogenesis, neuronal distribution and electrical activity in the neocortex [3–5]. Mutations in the *MEF2C* gene, including microdeletions, missense, or nonsense mutations, have been linked to a rare genetic disorder known as *MEF2C* haploinsufficiency syndrome. This syndrome is characterized by intellectual disability (ID), epilepsy, and additional autistic features like absent speech and impaired social interactions [6]. Genome-wide association studies (GWAS) have identified common variants in the *MEF2C* gene that are associated with schizophrenia (SCZ) intelligence (IQ) and educational attainment (EA) [7–9]. *MEF2C* is associated with genetic and epigenetic risk architectures of SCZ [10]. *MEF2C* motifs were present among the top-scoring single-nucleotide polymorphisms (SNPs) associated with SCZ in GWAS and deep sequencing of histone methylation landscapes in individuals with SCZ and controls revealed a significant abundance of *MEF2C* motifs associated with histone hypermethylation in the disorder. Additionally, the upregulation of *MEF2C* improved working memory, object recognition memory, and spinal remodeling in prefrontal projection neurons in mice [10].

Various studies have utilized *MEF2C* heterozygous or homozygous knockout (KO) animal models to investigate the role of *MEF2C* in brain function and to identify molecular mechanisms underlying human phenotypes associated with *MEF2C* [3,11–15]. Recently, Mohajeri *et al.* (2022) evaluated *MEF2C* loss-of-function mutations in human-derived cell lineages representing different stages of neural development [16]. They directly disrupted the gene by targeting the coding sequence of *MEF2C* with CRISPR-engineered mutations, resulting in 122kb and 131kb deletions of the gene. Expanding beyond direct disruption of the gene, they utilized an indirect approach to disrupt the 3D genome organization of the locus and the regulatory

architecture of the gene. Here, they either deleted the distal boundary (DB) or the proximal boundary (PB) of the topologically associated domain (TAD) encompassing *MEF2C*. Specifically, they performed a targeted deletion of a 3.3kb segment of the DB, which targeted a single occupied CTCF binding site located more than 1.3Mb distal to the *MEF2C* promoter. As for the PB, they carried out a targeted deletion of a single occupied CTCF binding site within a 3' intron of *MEF2C*. Following the direct or indirect mutation of *MEF2C* in human induced pluripotent stem cells (iPSCs), these cells were differentiated into neural stem cells (NSCs) and glutamatergic induced neurons (iNs) as cellular models [16]. NSCs are undifferentiated cells that have the ability to self-renew and generate various types of neurons and glial cells. Glutamatergic iNs are responsible for synthesizing glutamate, the primary excitatory neurotransmitter in the mammalian central nervous system. Glutamate plays a crucial role in various essential brain processes, including cognition, learning, memory, and sensory perception [17]. The study used these cellular models to investigate the impact of both direct and indirect disruptions of *MEF2C* on global transcriptional signatures and electrophysiological changes in human neurons [16]. Both the direct disruption and the loss of a PB, but not the deletion of a DB, led to down-regulation of *MEF2C* expression, which resulted in reduced synaptic activity. The presence of common differentially expressed genes (DEGs) associated with neurogenesis and neuronal differentiation in both direct and indirect *MEF2C* disruptions suggests shared functional consequences arising from both types of *MEF2C* disruption [16].

Here, we expanded upon the findings of Mohajeri *et al.* (2022) [16] by utilizing their gene expression data and combining it with chromatin immunoprecipitation sequencing (ChIP-seq) data for *MEF2C* (Fig 1) [18]. This integration enabled us to identify putative direct transcriptional targets of *MEF2C* in both NSCs and iNs that were dysregulated following either heterozygous or homozygous direct or indirect mutation of the gene. Given *MEF2C*'s association with neuropsychiatric disorders and cognitive function, we sought to investigate if the direct targets of *MEF2C* that are dysregulated by different mutations in the different cellular models are enriched for genes containing SNPs associated with SCZ and cognitive function from GWAS, as well as enriched for genes harboring rare *de novo* mutations (DNMs) contributing to neurodevelopmental disorders. Subsequently, we investigated the biological processes and specific cell types that are dysregulated as a consequence of *MEF2C* disruption to better understand the contribution of *MEF2C*-regulated genes to the molecular mechanisms of SCZ and cognition. Finally, we sought to identify trans-expression quantitative trait loci (trans-eQTL) at the *MEF2C* gene that are associated with altered expression of downstream *MEF2C* target genes (Fig 1).

## Materials and methods

### Ethics statement

Data were directly downloaded from published studies and no additional ethics approval was needed. Each study is referenced and details on ethics approval are available in each manuscript.

### *MEF2C* Transcriptomic data

We utilized transcriptomic data generated in a study of *MEF2C* conducted by Mohajeri *et al.* (2022) [16]. That study used dual-guide CRISPR-Cas9 genome editing to directly (deletion in the coding region) or indirectly (mutation of the PB of the TAD that encompasses *MEF2C*) disrupt *MEF2C* function in human iPSCs. Single cells were isolated and screened to identify edited clones and matched unedited controls. Six replicates per genotype (heterozygous (het) or homozygous (hom) deletion (DEL)) were then differentiated into NSCs and iNs and

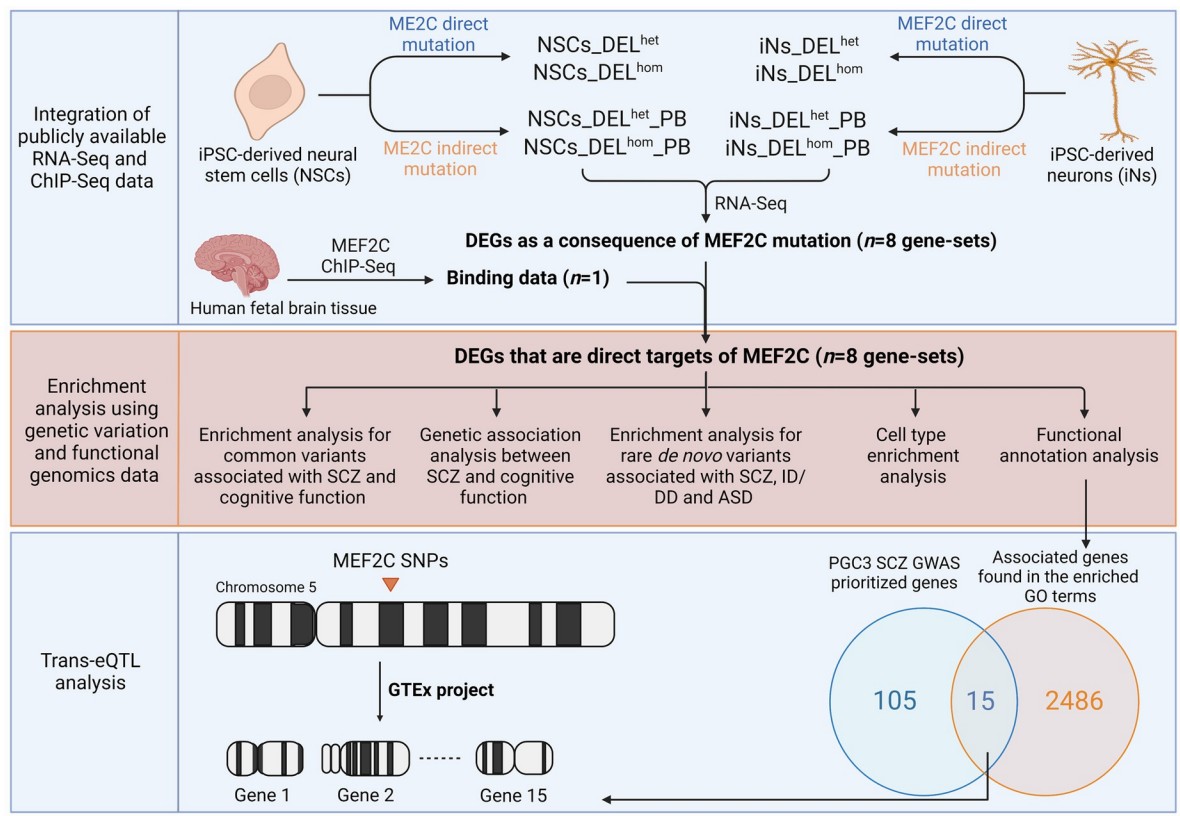

**Fig 1. Schematic representation illustrating the stepwise methodology employed in this study.** DELhom: Homozygous deletion; DELhet: Heterozygous deletion; PB: Proximal boundary (indirect mutation of MEF2C); DEGs: Differentially expressed genes; SCZ: Schizophrenia; ID: Intellectual disability; DD: Developmental delay; ASD: Autism spectrum disorder; GO: Gene ontology; PGC: Psychiatric Genomics Consortium; GWAS: Genome-wide association study; SNPs: Single nucleotide polymorphisms; eQTL: Expression quantitative trait loci; GTEx: Genotype-Tissue Expression. Figure created with BioRender.com.

subjected to RNA-seq analysis. This analysis generated eight sets of differentially expressed genes (DEGs) in these cellular models, labelled as follows with PB denoting the indirect mutation: NSCs_DEL$^{het}$, NSCs_DEL$^{hom}$, NSCs_DEL$^{het}$_PB, NSCs_DEL$^{hom}$_PB, iNs_DEL$^{het}$, iNs_DEL$^{hom}$, iNs_DEL$^{het}$_PB, and iNs_DEL$^{hom}$_PB. Each gene-set represented a different combination of *MEF2C* disruption and genotype in NSCs or iNs. The significant DEGs were identified at FDR < 0.1 (S1 Table).

## *MEF2C* ChIP-sequencing data analysis

To investigate *MEF2C* binding, we utilized existing ChIP-seq data from a study of *MEF2C* conducted on human fetal brain cultures [18]. Input DNA was used as a control. The raw files, comprising a single replicate, were provided by the authors. The quality of the raw FastQ files was assessed using the FastQC (http://bioinformatics.babraham.ac.uk/projects/fastqc/). Reads were aligned to the human genome (hg19) using Burrows Wheeler Aligner (BWA; http://bowtie-bio.sourceforge.net/bowtie2) [19]. Post-processing of the alignment data was conducted using Samtools (https://github.com/samtools). We converted the SAM files to BAM format, sorted the BAM files, removed any potential PCR duplicates and generated a file containing mapping statistics [20]. Peaks were called using MACS2 (parameters: -f BAM -g hg -q 0.01) [21]. ChIPSeeker was used to determine overlap with genomic features and for peak annotation to the nearest genes [22].

## Integrative Analysis of RNA-Seq and ChIP Data

To infer the direct target genes of *MEF2C*, the eight sets of DEGs described above were integrated with the *MEF2C* ChIP-seq data using the BETA (Binding and Expression Target Analysis) (http://cistrome.org/BETA/) package software [23]. BETA ranks genes based on two key factors: the regulatory potential of factor binding sites and the differential expression observed upon factor binding. The regulatory potential is assessed by considering the distance of the binding sites from the transcription start site and the cumulative impact of multiple binding sites. By considering both aspects, a rank product (RP) was calculated for each gene, which can be interpreted as a probability, indicating the likelihood that a gene is a true direct target of *MEF2C* based on both criteria. Genes with a conservative RP < 0.01 were considered as direct targets of *MEF2C* (S2 Table).

## Stratified linkage disequilibrium score regression (sLDSC) Analysis

Stratified linkage disequilibrium score regression (sLDSC) (https://github.com/bulik/ldsc) [24] was used to investigate if the *MEF2C* direct targets were enriched for heritability contributing to SCZ, IQ, and EA phenotypes. GWAS summary statistics for these phenotypes [7–9] were obtained from publicly available databases (the Psychiatric Genomics Consortium Website www.med.unc.edu/pgc, the Complex Trait Genetics lab www.ctg.cncr.nl/, and the Social Science Genetic Association Consortium www.thessgac.org/data). For control purposes, we performed sLDSC analysis using GWAS summary statistics for an additional four phenotypes, including attention deficit hyperactivity disorder (ADHD) [25], obsessive–compulsive disorder (OCD)[26], Alzheimer's disease (AD) [27] and stroke [28]. Linkage disequilibrium (LD) scores between SNPs were estimated using the 1000 Genomes Phase 3 European reference panel. SNPs present in HapMap 3 with an allele frequency > 0.05 were included. Enrichment of heritability was assessed controlling for the effects of 53 functional annotations included in the full baseline model version. Enrichment for heritability was compared to the baseline model using the Z-score to derive a (one-tailed) P-value. Significant enrichments were determined using a Bonferroni correction, which set the corrected P value threshold at < 2.08E-03.

## Overlapping Genes Implicated in the GWAS of SCZ and IQ/EA

In the GWAS of IQ, 1,016 genes were reported as associated with IQ through positional mapping, eQTL mapping, chromatin interaction mapping and gene-based association analysis [8]. For EA, 1,838 genes were prioritized using Data-driven Expression *Prioritized* Integration for Complex Traits (DEPICT), which was based on correlations across reconstituted gene-sets [9]. For SCZ, 682 associated genes were identified through fine mapping and summary-data-based mendelian randomization [7] (S3 Table). To investigate distinct and overlapping associations with SCZ and cognitive function, we identified genes that are associated with SCZ but not IQ or EA (n = 472), genes that are associated with at least one of the cognitive phenotypes (IQ or EA) but not SCZ (n = 2,258) and genes that are associated with both SCZ and at least one of the cognitive phenotypes (IQ or EA; n = 210).

## Gene-set based polygenic risk score (PRS)

PRSice-2 software [20] was utilized for gene-set based PRS analysis aiming to investigate whether the *MEF2C* target gene-sets contributed to the shared genetic basis of SCZ and cognitive traits. PRSice-2 calculates PRS for each individual by summing up the number of minor alleles at each SNP multiplied by the GWAS effect size. It performs regression analysis, adjusting for sex, age, and GWAS array type as covariates, and provides performance metrics

(Nagelkerke's R2 and P value). SNP P values and effect sizes for SCZ were derived from a SCZ GWAS meta-analysis on 40,675 cases and 64,643 controls [7]. Irish samples were excluded from this GWAS to keep that base/discovery sample independent from the target/test sample for the PRS analysis. The SNP P values and effect sizes for IQs were based on an IQ GWAS on 269,867 individuals [8]. For each gene-set, SNPs in high LD were clumped according to PRSice-2 guidelines. Genotype data for the identified SNPs were extracted from the full GWAS data of the Irish samples, which consisted of 1,512 individuals, including SCZ patients and controls with IQ measurements [29,30]. Effect-size weighted SCZ-PRS and IQ-PRS were computed for each gene-set using thresholds ranging from P < 0.05 to P≤1 (*P < 0.05*, 0.1, 0.15, 0.2, *1*). To validate the findings, 10,000 randomized phenotypes (equally distributed cases and controls as per the original dataset) were generated from the Irish samples, and SCZ-PRS and IQ-PRS were calculated for each gene-set using the randomized data to obtain empirical P values.

### Analysis of *De Novo* Mutations

The R package denovolyzeR (http://denovolyzer.org/) was used to test for enrichment of rare *de novo* mutations (DNMs) in our gene-sets, estimating the expected number of DNMs for each gene based on sequence context and gene size [31]. Synonymous (Syn), missense (Mis), and loss of function (Lof) (including nonsense, frameshift, and splice) DNMs reported in exome sequencing studies of SCZ (*n* = 3447 trios) [32–35], ASD (*n* = 6430 trios) [36], and ID and/or DD (*n* = 4,851 trios) [37–40] and unaffected siblings (*n* = 1,995) [32]. S4 Table provides details about each study along with the respective table names listing the identified DNMs. To ensure consistency with the Deciphering Developmental Disorders Study (2017), we applied a filtering step for DNMs. Specifically, DNMs with a posterior probability score below 0.00781 were excluded. Enrichment of DNMs in a gene-set was investigated using a two-sample Poisson rate ratio test, using the ratio of observed to expected DNMs in genes outside of the gene-set as a background model. Significant enrichments were determined using a Bonferroni correction, which set the corrected P value threshold at < 5.21E-04.

### *MEF2C* Direct Target Genes Analysis with Single-cell RNA-seq

The Expression Weighted Cell-type Enrichment (EWCE) R package (https://github.com/NathanSkene/EWCE) was used to assess if the direct target genes of *MEF2C* had higher expression in a particular cell type than expected by chance [41]. This method generates random gene sets (*n* = 100,000) of equal length from background genes to estimate the probability distribution. We performed enrichment analysis in a prenatal human dataset and in an adult human dataset. The prenatal human dataset included single-nuclei RNA sequencing (snRNA-seq) data from three fetuses from the second trimester of gestation and contained data for 91 distinct clusters of nuclei from five brain regions (frontal cortex (FC), ganglionic eminence (GE), hippocampus (Hipp), thalamus (Thal), and cerebellum (Cer)) [42]. The adult human dataset included data for 120 distinct clusters of nuclei from across the human cortex covering the middle temporal gyrus (MTG), cingulate gyrus (CgG), primary visual cortex (V1C), primary somatosensory cortex (S1C) and the primary motor cortex (M1C). Nuclei were sampled from postmortem and neurosurgical (MTG only) donor brains (https://portal.brain-map.org/atlases-and-data/rnaseq/protocols-human-cortex) [43,44]. The significance of the enriched expression of the *MEF2C* direct target genes relative to the background genes in each cell type was assessed by calculating the difference in standard deviations between the two expression profiles. Statistically significant enrichments were determined using a Bonferroni correction to adjust for multiple testing of cell types.

### Functional enrichment analysis

ClueGO (version 2.5.9), a plugin for Cytoscape (version 3.8.2) was used to identify the Gene Ontology (GO) terms (categorized as biological processes (BP), molecular functions (MF), and cellular components (CC)) and the biological pathways (KEGG, Reactome, WikiPathways) enriched for (i) genes proximal to MEF2C peaks identified via ChIP-seq analysis using brain tissue-expressed genes as the background gene-set (S5 Table) [45] and (ii) the *MEF2C* direct target gene-sets using specific cell-type expressed genes as the background gene-set. Brain tissue-expressed genes were obtained directly from the Human Protein Atlas database (https://www.proteinatlas.org/) [46]. Cell type-specific expressed genes were identified by calculating Transcripts Per Million (TPM) values from raw reads counts of wild type NSCs and iNs downloaded from the gene expression omnibus (GEO GSE204778). Genes with TPM values less than 1 were filtered out. GO term relationships were determined based on shared genes and assessed using chance-corrected kappa statistics. Bonferroni correction was applied to adjust for multiple testing.

## Results

### Identification and Annotation of MEF2C Binding Peaks

Analysis of ChIP-Seq data using MACS and ChIPSeeker identified 10,620 MEF2C binding peak regions (FDR $\leq$ 1%) across the entire genome. Approximately 80% of the peaks were located in close proximity to gene-encoding regions including promoters (< = 1-kb (55.8%), 1–2 kb (2.35%), 2–3 kb (1.99%)), 5' UTRs and 3' UTRs (0.44%), exons (0.18%), first introns (5.8%), and other intron regions (14.8%) (S6 Table). When the binding peaks were mapped to the closest RefSeq annotated transcripts, they were found in close proximity to 5,775 protein coding genes. GO annotation analysis showed these genes were most involved in RNA binding, transcription and functions within the nucleus (S7 Table).

### Identification of *MEF2C* Direct Target Genes

We integrated *MEF2C* ChIP-seq data with data on DEGs from cell line models where *MEF2C* had been mutated (2 cell types (NSCs or iNs) x 2 DEL mutation types (direct or indirect (PB)) x 2 genotypes (het or hom) = 8 cell line models). Fig 1 illustrates the stepwise methodology employed in this study, from generating the eight gene-sets from these models through to enrichment analysis using genetic variation and functional genomics data. We identified that *MEF2C* had a direct regulatory influence on approximately 23–57% of the DEGs from the original study (Table 1). Shared *MEF2C* direct target genes were observed between different genotypes within the same cell type in both NSCs and iNs. In NSCs, the proportion of shared genes for both heterozygous and homozygous genotypes ranged from 37% (direct mutation) to 42% (indirect mutation), while in iNs, it ranged from 23% (direct mutation) to 34% (indirect mutation) (S1 Fig). Furthermore, there was a limited number of common *MEF2C* direct target genes found for the same genotype across different gene disruption types, ranging from 13% (homozygous) to 22% (heterozygous) in NSCs and from 8% (homozygous) to 18% (heterozygous) in iNs (S1 Fig). A small fraction of the *MEF2C* direct target genes (3% in NSCs and 1.4% in iNs) were in each gene-set (S1 Fig), indicating that the downstream effect of different gene mutations and their genotypic state is to mostly impact distinct sets of genes.

### Enrichment analysis for genes containing common variants

We performed sLDSC analysis to investigate if the *MEF2C* direct target gene-sets are enriched for genes containing common genetic variants associated with SCZ risk or cognitive ability [7–

**Table 1. Number of DEGs in each gene-set before and after the integration with MEF2C ChIP-seq data to identify direct target genes.**

| Gene-Set | # of all DEGs | # of *MEF2C* Direct Targets (% of MEF2C Direct Targets relative to all DEGs) |
|---|---|---|
| NSCs | | |
| DEL^het | 366 | 169 (46%) |
| DEL^hom | 2187 | 1170 (53%) |
| DEL^het_PB | 492 | 240 (49%) |
| DEL^hom _PB | 728 | 412 (57%) |
| iNs | | |
| DEL^het | 689 | 335 (51%) |
| DEL^hom | 291 | 145 (50%) |
| DEL^het_PB | 2980 | 1034 (35%) |
| DEL^hom _PB | 5132 | 1174 (23%) |

NSCs: Neural stem cells; iNs: Induced neurons; DEL^hom: Homozygous deletion; DEL^het: Heterozygous deletion; PB: Proximal boundary (indirect mutation of MEF2C).

9]. Six of the eight gene-sets were significantly enriched for heritability contributing to at least one of the studied phenotypes (SCZ, IQ, and/or EA) (Fig 2). Specifically, three gene-sets (NSCs_DEL^hom, iNs_DEL^het_PB, and iNs_DEL^hom_PB) were significantly enriched for all phenotypes after multiple testing correction (Fig 2 and S8 Table). These findings highlight the potential role of *MEF2C* in regulating genes involved in SCZ and cognitive function. When we

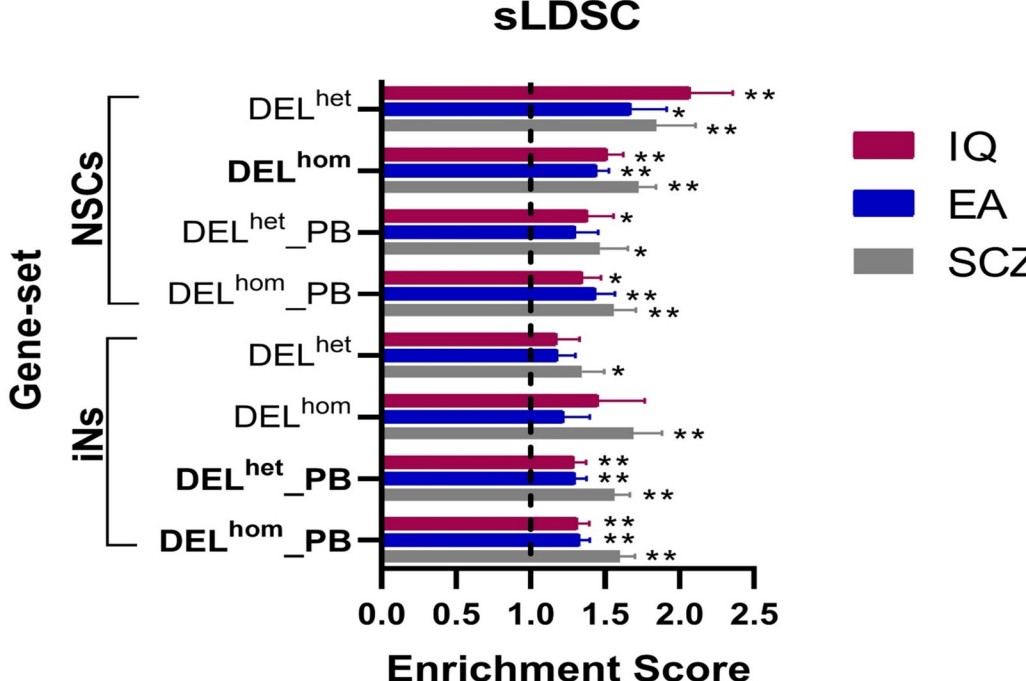

**Fig 2. Results from sLDSC analysis of MEF2C direct target gene-sets using GWAS data.** The graph plots the enrichment values, defined as the ratio of heritability (h2) to the number SNPs, on the x-axis. The error bars represent the standard errors. Two asterisks (**) indicate significance after Bonferroni correction, and one asterisk (*) indicates nominal significance. Gene-sets enriched for the three phenotypes are highlighted in bold. NSCs: Neural stem cells; iNs: induced neurons; DELhom: Homozygous deletion; DELhet: Heterozygous deletion; PB: Proximal Boundary.

removed genes associated with SCZ from the enrichment analysis of IQ and EA, we saw that the majority of enriched gene-sets remained significant. When we removed genes associated with IQ or EA from the enrichment analysis of SCZ, we saw that only the enrichment of the NSCs_DEL$^{hom}$ gene-set remained significant (S9 Table). This suggests that some *MEF2C* target genes are contributing to both SCZ and cognitive phenotypes while others are more phenotype specific. No significant enrichment was detected for any of the four control phenotypes (S10 Table).

### Genetic Association between SCZ and cognitive function

To explore the genetic overlap between these phenotypes further, gene-set based PRS analysis was conducted to investigate if the *MEF2C* target gene-sets contributed to the shared genetic etiology between SCZ and cognition. This was done by generating a gene-set PRS based on SCZ risk from GWAS and testing if this SCZ-PRS could explain variance in IQ in an independent dataset. We also tested if a gene-set IQ-PRS could predict SCZ case-control status in independent dataset. While the IQ-PRS could not predict SCZ case-control status, we found that the SCZ-PRS derived from three of the eight gene-sets (NSCs_DEL$^{hom}$, iNs_DEL$^{het}$_PB, and iNs_DEL$^{hom}$_PB) could explain a significant proportion of variance in IQ (Fig 3 and S11 Table). These are the same three gene-sets that were previously enriched for common variation

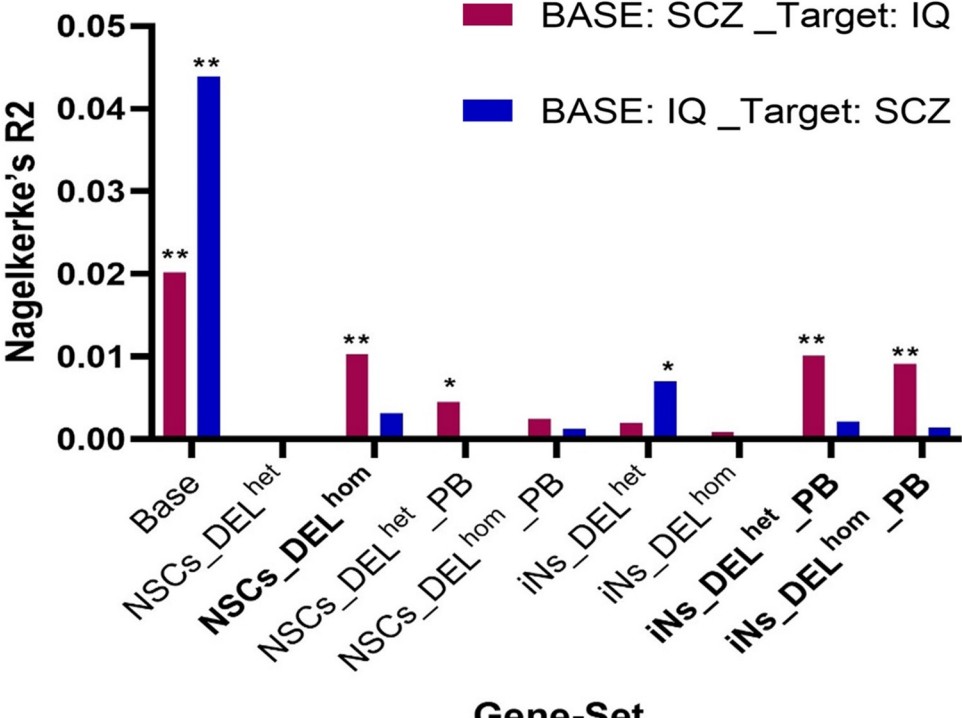

**Fig 3. Gene-set based PRS analysis to examine associations between the SCZ-PRS and IQ, and between the IQ-PRS and SCZ.** Base on the x-axis refers to a PRS generated using all variants in the genome. The height of the columns on the y-axis indicates the proportion of variance in the phenotype explained when a gene-set based PRS is constructed using SCZ GWAS data and is tested against IQ (pink columns) or when a gene-set based PRS is constructed using IQ GWAS data and is tested against SCZ case-control status (blue columns). Two asterisks (**) indicate significance after Bonferroni correction, and one asterisk (*) indicates nominal significance. NSCs: Neural stem cells; iNs: induced neurons; DELhom: Homozygous deletion; DELhet: Heterozygous deletion; PB: Proximal Boundary.

associated with SCZ, IQ and EA. When we removed genes associated with IQ or EA from these gene-sets, these three SCZ-PRSs could still explain variance in IQ in an independent dataset at levels that were nominally significant (S12 Table). These findings suggest that genetic variants associated with SCZ within these gene-sets also influence cognitive performance and the effect was not just due to genes already associated with IQ or EA. We performed a sensitivity analysis with respect to the p-value threshold for SNP inclusion and found that results were consistent and stable across different p-value thresholds (S11 and S12 Tables).

### Enrichment Analysis for Genes Containing *De Novo* Mutations

To assess the impact of rare variants within the *MEF2C* gene-sets on SCZ and other neurodevelopmental disorders where cognitive impairment is a major feature (ASD, ID and DD), we examined whether these gene-sets exhibited enrichment for Syn, Mis, and Lof DNMs in trio-based exome sequencing studies of these disorders [32–40]. The same gene-sets that showed enrichment for common variants associated with SZ, IQ and EA (NSCs_DEL$^{hom}$, iNs_DEL$^{het}$_PB, and iNs_DEL$^{hom}$_PB) were also significantly enriched for genes containing Lof and/or Mis DNMs reported specifically in ID and/or DD patients after multiple test correction (Table 2). The NSCs_DEL$^{hom}$ gene-set was also significantly enriched for Lof DNMs found in people with autism (Tables 2 and S13). None of our gene-sets showed enrichment for genes containing rare DNMs reported in SCZ patients. As a control, our gene-sets were not enriched for Syn DNMs reported for these disorders and not enriched for any class of DNM reported in the unaffected siblings of patients.

### Cell-type enrichment analysis

We utilized the EWCE R package [41] to investigate which individual cell types are enriched for these genes in the prenatal and adult human brain using snRNA-seq data. The three gene-sets with by far the most enriched cell types are the three gene-sets that were enriched for common variation associated with SCZ, IQ and EA and rare DNMs reported in neurodevelopmental disorders (NSCs_DEL$^{hom}$, iNs_DEL$^{het}$_PB and iNs_DEL$^{hom}$_PB). There is a consistent pattern for these gene-sets in the prenatal and adult data with both glutamatergic excitatory

**Table 2. Rare variant enrichment analysis of MEF2C direct target gene-sets using data on DNMs, identified in patients with SCZ, ASD, ID and DD.**

| Gene-Set | SCZ<br>*n* = 3394 trios | ASD<br>*n* = 6430 trios | ID/DD<br>*n* = 4485 trios | Unaffected Siblings<br>*n* = 1995 |
|---|---|---|---|---|
| **NSCs** | | | | |
| DEL$^{het}$ | ns | ns | ns | ns |
| DEL$^{hom}$ | ns | **Lof**\*\* | **Mis**\*\*, **Lof**\*\* | ns |
| DEL$^{het}$_PB | ns | Lof\* | ns | ns |
| DEL$^{hom}$_PB | ns | Lof\* | Mis | ns |
| **iNs** | | | | |
| DEL$^{het}$ | ns | ns | Mis\* | ns |
| DEL$^{hom}$ | Syn\* | ns | Mis\*, Lof\* | ns |
| DEL$^{het}$_PB | ns | ns | **Mis**\*\* | ns |
| DEL$^{hom}$_PB | Lof\* | Mis\*, Lof\* | **Mis**\*\*, **Lof**\*\* | ns |

Two asterisks (\*\*) indicate significant enrichment for mutation type after Bonferroni correction, one asterisk (\*) indicates significant enrichment for mutation type at nominal significance level and ns indicates non-significant for all classes of mutation tested. SCZ: Schizophrenia; ASD: Autism spectrum disorder; ID: Intellectual disability; DD: Developmental delay; Syn: Synonymous mutations, Mis: Missense mutations; Lof: Loss-of-function mutations; NSCs: Neural stem cells; iNs: Induced neurons; DEL$^{hom}$: Homozygous deletion; DEL$^{het}$: Heterozygous deletion; PB: Proximal boundary (indirect mutation of MEF2C).

neurons and GABAergic inhibitory neurons enriched across different regions of the prenatal brain and across regions of the adult cortex (S14 and S15 Tables). The enrichment of genes in both excitatory and inhibitory neurons is consistent with the role of *MEF2C* in regulating the balance of excitatory and inhibitory synapses, the disruption of which may contribute to neurodevelopmental disease [11]. Lastly, the NSCs_DEL[hom] gene-set was enriched within cycling progenitor cells and intermediate progenitor cells within the prenatal brain, which like the NSCs can produce new types of neurons and glial cells (S14 Table).

### Functional enrichment analysis

ClueGO (version 2.5.9), a plugin for Cytoscape (version 3.8.2) was used to investigate if genes within the eight sets are over-represented in similar or distinct GO terms for biological processes, cellular components and molecular functions, and biological pathways, using specific cell-type expressed genes as the background gene-set. The NSCs gene-sets were enriched for GO terms related to neuron development, regulation of neuron and glial cell differentiation and regulation of metabolic processes (Figs 4–6 and S16–S19 Tables). The iNs gene-sets were

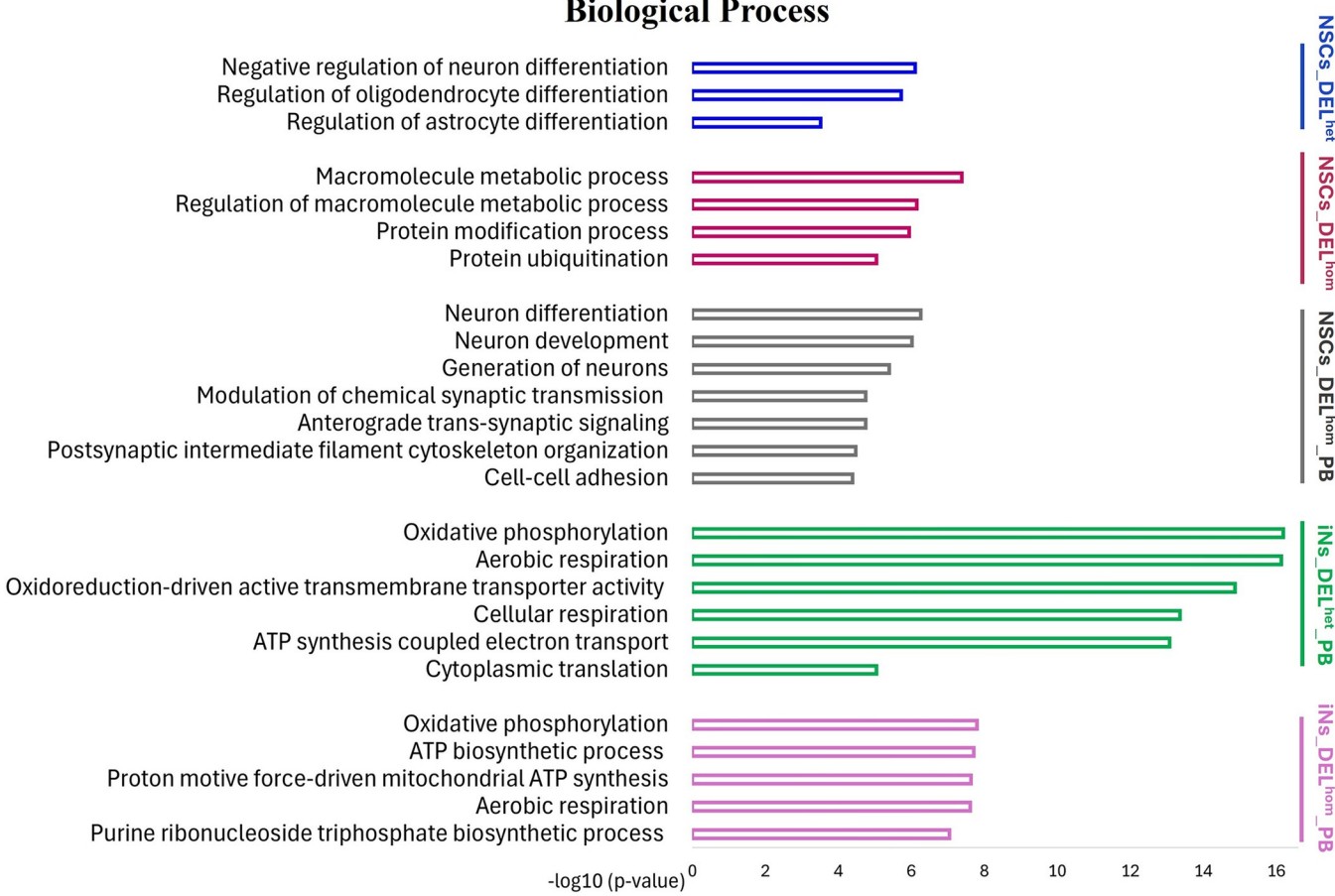

**Fig 4. Bar charts of gene ontology (GO) analysis of biological process for MEF2C direct target gene-sets using the ClueGo plugins of Cytoscape.** The Bonferroni method was applied for a p-value correlation (p < 0.05). The vertical axis displays the names of the GO terms. The horizontal axis and bar lengths represent the significance [−log10 (p-value)]. Colors in the bars represent different MEF2C direct target gene-sets. Results are presented only for the five gene-sets that were previously enriched for common variation associated with SCZ, IQ and/or EA. Enriched terms that were related to each other in the ontology were grouped together, with the most significant term(s)/group displayed. All data is detailed in S16–S23 Tables. NSCs: Neural stem cells; iNs: induced neurons; DELhom: Homozygous deletion; DELhet: Heterozygous deletion; PB: Proximal Boundary.

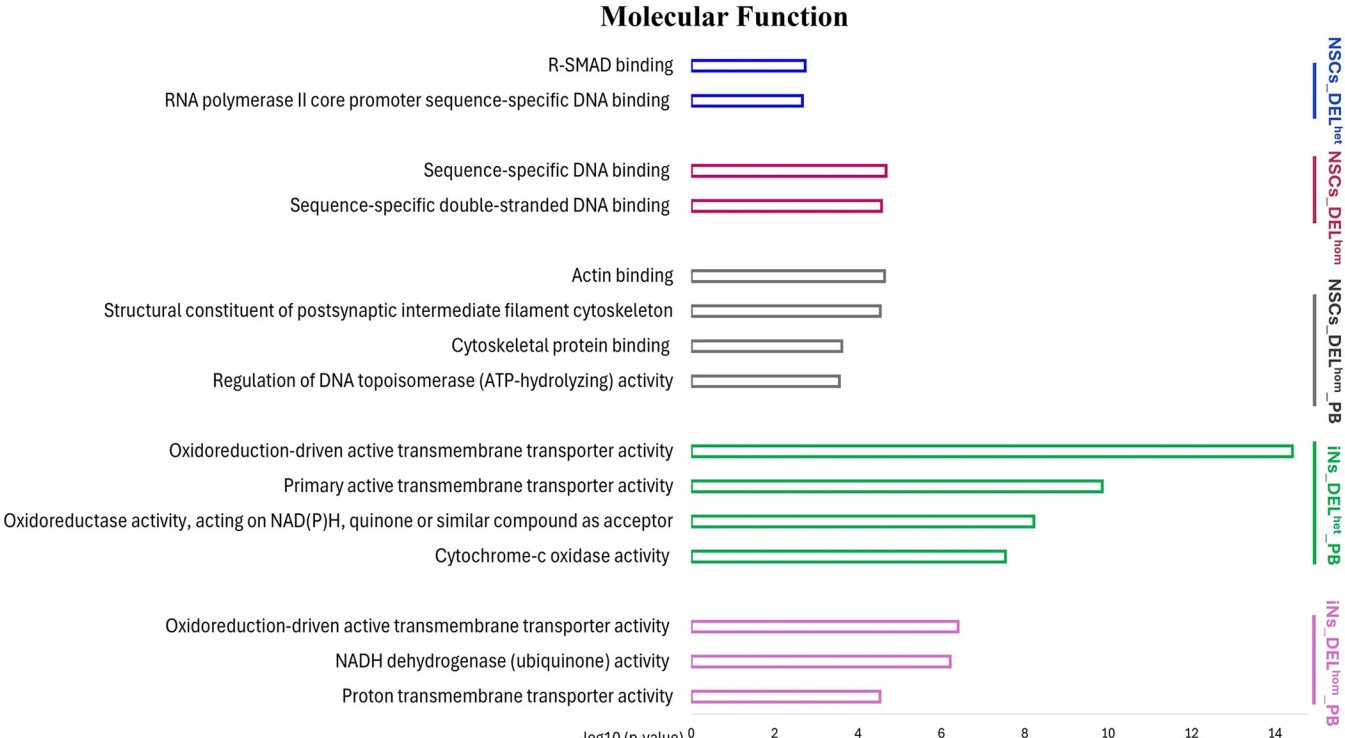

**Fig 5. Bar charts of gene ontology (GO) analysis of molecular function for MEF2C direct target gene-sets using the ClueGo plugins of Cytoscape.** The Bonferroni method was applied for a p-value correlation (p < 0.05). The vertical axis displays the names of the GO terms. The horizontal axis and bar lengths represent the significance [−log10 (p-value)]. Colors in the bars represent different MEF2C direct target gene-sets. Results are presented only for the five gene-sets that were previously enriched for common variation associated with SCZ, IQ and/or EA. Enriched terms that were related to each other in the ontology were grouped together, with the most significant term(s)/group displayed. All data is detailed in S16–S23 Tables. NSCs: Neural stem cells; iNs: induced neurons; DELhom: Homozygous deletion; DELhet: Heterozygous deletion; PB: Proximal Boundary.

enriched for GO terms related to mitochondrial function and energy production, including the oxidative phosphorylation process (Figs 4–6 and S20–S23 Tables). KEGG pathway analysis revealed that the NSCs gene-sets were enriched in pathways including *Orexin receptor pathway*, *Protein processing in endoplasmic reticulum and Aerobic glycolysis*, while the iNs gene-sets were enriched in pathways including *The citric acid (TCA) cycle and respiratory electron transport* and *Oxidative phosphorylation* (S16–S23 Tables). The enriched GO terms following disruption of MEF2C are distinct from those observed earlier from the ChIP-seq binding pattern of MEF2C in the absence of any gene disruption.

## Trans expression quantitative trait loci analysis

We hypothesized that genetic variation at *MEF2C* (associated with SCZ, IQ or EA) could indirectly affect expression of a downstream target gene, mediated through MEF2C's role as a transcription factor. This would be a trans expression quantitative trait loci (eQTL) effect and evidence of two risk genes (i.e., *MEF2C* and a downstream target gene) functioning within a putative risk pathway. To reduce the number of possible tests of target genes, we first restricted this analysis to the five gene-sets that were enriched for association with at least one of SCZ, IQ or EA. We next limited these *MEF2C* direct target genes to only those in significantly enriched GO terms (to capture genes with relevant functions; S24 Table) and to those genes among the 120 genes prioritized in the latest GWAS for SCZ (S25 Table) [7]. These 120 genes were identified through a combination of fine-mapping, transcriptomic analysis and functional genomic

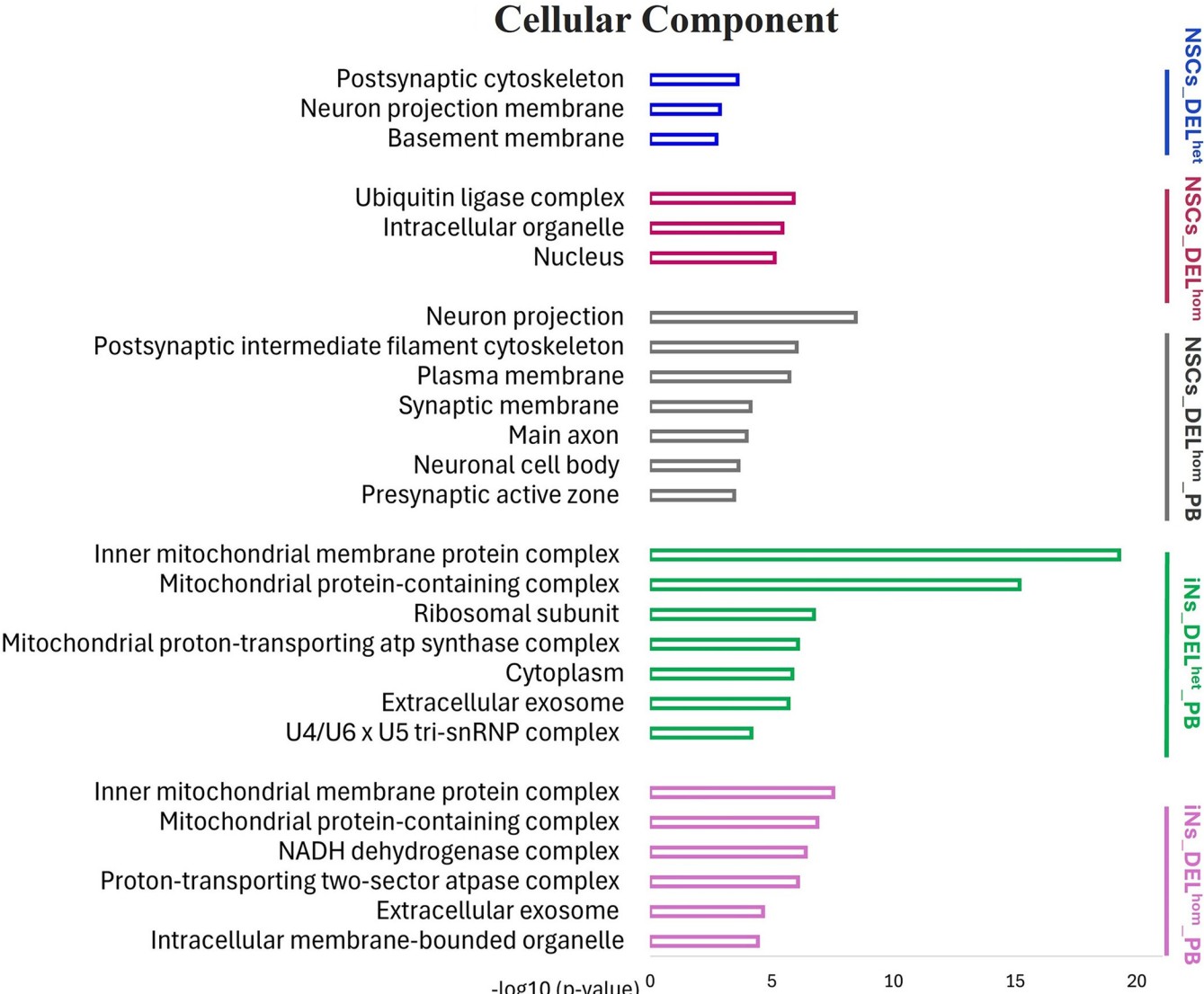

**Fig 6. Bar charts of gene ontology (GO) analysis of cellular component for MEF2C direct target gene-sets using the ClueGo plugins of Cytoscape.** The Bonferroni method was applied for a p-value correlation (p < 0.05). The vertical axis displays the names of the GO terms. The horizontal axis and bar lengths represent the significance [−log10 (p-value)]. Colors in the bars represent different MEF2C direct target gene-sets. Results are presented only for the five gene-sets that were previously enriched for common variation associated with SCZ, IQ and/or EA. Enriched terms that were related to each other in the ontology were grouped together, with the most significant term(s)/group displayed. All data is detailed in S16–S23 Tables. NSCs: Neural stem cells; iNs: induced neurons; DELhom: Homozygous deletion; DELhet: Heterozygous deletion; PB: Proximal Boundary.

annotations [7]. The GWASs of IQ and EA had not performed similar prioritization analysis and each reported >1,000 associated genes. Fifteen of the 120 genes are *MEF2C* direct target genes that were in the enriched GO terms (S26 Table). We took 10 LD-independent SNPs at *MEF2C* that were associated with SCZ, EA, and IQ at genome-wide significant levels (S27 Table) and investigated their association with the expression levels of these fifteen genes using eQTL data obtained from the Genotype-Tissue Expression (GTEx) project (https://gtexportal.org/home/) [47]. We detected a trans eQTL for a single SNP at *MEF2C* (rs6893807; associated with IQ in GWAS) on the expression of the SCZ risk gene *BNIP3L* in the cerebellar hemisphere following multiple testing correction (P = 1.60E-05, adjusted P = 0.025). The *BNIP3L* gene is known to be involved in mitophagy, a process responsible for the selective removal of

damaged mitochondria. Finally, to further explore genes with mitochondrial functions beyond the 120 prioritized SCZ genes, we performed a second trans eQTL analysis. This time we restricted the target genes to those within enriched GO terms related to mitochondrial function and energy production that had cell-type specific expression in the enriched cell-types from that earlier analysis. As a result, we tested the 10 LD-independent SNPs at MEF2C against 300 genes. Overall, the trans eQTL effect on the expression of *BNIP3L*, already detected, was the only finding that survived multiple test correction (S28 Table).

## Discussion

The present study aimed to integrate transcriptomic data from human neural cell models of *MEF2C* deletion with ChIP-seq data to identify the direct regulatory influence of *MEF2C* disruption on global transcriptional signatures. These data from models of early neuronal development stem cells (NSCs) and fully differentiated neurons (iNs) provide insight into the sets of genes downstream of *MEF2C* that may be important for brain function at different stages of neurodevelopment. Common variants in *MEF2C* are associated with SCZ and cognitive function. We do not expect these sets of downstream dysregulated genes to directly align with the molecular mechanisms of SCZ and cognitive function. But we have been able to interrogate these gene sets to determine if they were enriched for other genes associated with these phenotypes or other neurodevelopmental disorders. From there, we investigated the functionality of the genes within these sets to generate evidence that supports existing hypotheses about the molecular basis of SCZ.

All eight gene-sets were significantly enriched for genes associated with at least one phenotype but three gene-sets (NSCs_DELhom, iNs_DELhet_PB, and iNs_DELhom_PB) were enriched for common variants associated with SCZ, IQ and EA and were further enriched for rare DNMs reported in ID and/or DD patients. We also showed using PRS analysis that genetic risk for SCZ in each of these gene-sets could explain a significant proportion of variance in IQ. These data support a role for the genes in these sets in the aetiology of SCZ risk and associated cognitive dysfunction. Functional enrichment analysis has indicated that genes regulated by *MEF2C* may have a dual function in neurodevelopment. In the early stages, they are implicated in neuron generation, differentiation and development, and metabolic processes, while in later stages, these genes are involved in mitochondrial function and energy production.

The process of neurogenesis forms the fundamental basis of brain development, involving the differentiation of NSCs and neural progenitor cells (NPCs) into mature neurons [48]. NSCs have the capacity to differentiate into various functional neural lineage cells, such as neurons, astrocytes, and oligodendrocytes [49]. Aberrant neurogenesis from NSCs has been implicated as a potential underlying mechanism in the development of neuropsychiatric disorders [50,51]. This critical process appears to be susceptible to various genetic and environmental disruptions during early brain development. The cell-type enrichment analysis of our NSCs gene-sets indicated that their constituent genes are enriched within cycling progenitor cells and intermediate progenitor cells in the prenatal brain, which can produce new types of neurons and glial cells. GO analysis of the NSCs gene-sets also indicated a role for *MEF2C*-regulated genes in neuron development and differentiation. These findings suggest that the differentiation process from NSCs to these specific neuronal subtypes may be influenced by *MEF2C* disruption, and variants within the genes that encode this process may contribute to SCZ risk and cognitive dysfunction. Dysregulation in the normal development and functioning of these neural lineage cells and imbalance between them have been strongly linked to the underlying causes of SCZ and other neuropsychiatric disorders [52,53]. Enrichment analysis

of KEGG pathways identified an enrichment of MEF2C direct target genes in NSCs within *Orexin receptor pathway*. The regulatory role of orexin (OXA) extends to various functions including sleep-wake rhythms, attention, cognition, and energy balance, all of which exhibit significant alterations in individuals with SCZ. Research has found inconsistent links between the OXA system and SCZ. Schizophrenia patients show decreased OXA plasma levels and hypothalamic OXA, with lower cortical OX2R mRNA in females. Conversely, males exhibit higher cortical OX1R and OX2R mRNA levels [54]. Furthermore, elevated OXA plasma levels have been associated with negative and disorganized symptoms in some studies [55]. We also observed that both glutamatergic excitatory neurons and GABAergic inhibitory interneurons in the prenatal and adult brain were enriched for genes from this NSCs set and from the two iNs sets. This provides further support for the balance of excitatory and inhibitory synapses, which is affected by *MEF2C* disruption [11], representing a potential molecular mechanism for neurodevelopmental disorders.

Synaptic activity is known to be an energy-intensive process that relies heavily on adenosine triphosphate (ATP) produced through oxidative phosphorylation (OXPHOS) in mitochondria [56]. OXPHOS involves the activity of electron transport chain (ETC) complexes (Complex I, II, III, and IV) and ATP synthase (Complex V), where electrons produced by the citric acid cycle are transferred across mitochondrial respiratory complexes [57]. Mitochondrial ATP production is crucial for various neuronal functions, including the assembly of the actin cytoskeleton for growth cone formation, development of pre-synaptic compartments, generation of membrane potential, and synaptic vesicle recycling and endocytosis. These processes contribute to essential synaptic activities and neuronal communication [58–60]. GO analysis of the iNs gene-sets identified that *MEF2C* directly regulates genes involved in ATP production, including those associated with OXPHOS.

Mitochondrial dysfunction has been implicated in the complex genetic mechanisms underlying SCZ. A total of 295 mitochondria-related genes associated with SCZ were identified through the examination of various studies encompassing copy number variants (CNVs), rare and *de novo* mutations, genome-wide associated SNPs, transcriptomic and proteomic studies of brain tissue from SCZ patients (reviewed in [61]). Significant associations were identified between SCZ and 19 nuclear mitochondria-related genes using GWAS data [62]. Four of these genes (*SMDT1*, *HSPE1*, *COQ10B*, and *FOXO3*) are in our iNs gene-sets. *FOXO3*, which is also significantly associated with IQ [8] is a transcription factor. It can translocate to the mitochondria where it may bind to mtDNA and react with mitochondrial transcription factor A (*TFAM*) and mitochondrial RNA polymerase (mtRNApol), inducing the production of various mitochondrial genes necessary for OXPHOS [63]. Large-scale brain eQTL studies have shown significant enrichment of mitochondria-related genes. Approximately 28% of the eQTL genes implicated in SCZ were related to mitochondria [64]. Furthermore, studies investigating gene expression in postmortem brain tissues of individuals with SCZ have consistently revealed a reduction in the expression of mitochondria-related genes. Specifically, genes such as NADH: ubiquinone oxidoreductase core subunit V1 (*NDUFV1*), NADH: ubiquinone oxidoreductase core subunit V2 (*NDUFV2*), NADH: ubiquinone oxidoreductase core subunit S1 (*NDUFS1*) [65–67], and cytochrome c oxidase (*COX*) show decreased expression levels in a region-specific manner [68]. *NDUFV2* and multiple isoforms of COX are present in the iNs gene-sets.

One of the most well established CNVs associated with SCZ is the deletion of chromosome 22q11.2, also known as 22q11.2 deletion syndrome (22q11.2DS) [69]. Individuals with 22q11DS often encounter cognitive impairments and a variety of neuropsychiatric disorders, including attention deficit hyperactivity disorder (ADHD), SCZ, anxiety, and ASD [69]. Among the genes deleted in 22q11DS, six (*MRPL40*, *PRODH*, *SLC25A1*, *TXNRD2*, *T10*, and

*ZDHHC8*) encode for mitochondrial proteins, and 3 others (*COMT*, *UFD1L*, and *DGCR8*) have an indirect effect on mitochondrial function [70]. A recent study demonstrated mito-chondrial deficits in iPSC-derived neurons from individuals with 22q11DS and SCZ. These deficits included reduced ATP levels, impaired activity of ETC complexes I and IV, and decreased levels of mitochondrial-translated proteins [71]. Our study revealed the direct regu-latory influence of *MEF2C* on two mitochondrial-related genes (*TXNRD2* and *COMT*) located within the 22q11.2 region in iNs. *TXNRD2* encodes for the mitochondrial Thioredoxin Reduc-tase 2, an enzyme that is essential for reactive oxygen species clearance in brain. In 22q model transgenic mice, mitochondrial TXNRD2 has been shown to impact synaptic function and is associated with long-range cortical connectivity and psychosis-related cognitive deficits [72]. A recent investigation demonstrated that an amyotrophic lateral sclerosis (ALS)-associated SNP located in the intronic region of MEF2C (rs304152), residing in a putative enhancer ele-ment, causes neuronal mitochondrial dysfunction. This dysfunction is characterized by decreased mitochondrial gene expression, impaired ATP production, increased oxidative stress, and decreased mitochondrial membrane potential [73]. Additionally, mitochondrial dysfunction can contribute to an imbalance in the excitatory (glutamate) and inhibitory (GABA) neurotransmitter systems [74–76], which we have referenced already as a potential molecular mechanism of neurodevelopmental disorder.

The most recent GWAS of SCZ prioritized 120 genes from the 287 genome-wide significant loci [7]. We identified a trans eQTL effect of a SNP in *MEF2C* on the expression of one of these prioritized genes, *BNIP3L*. Disruption of *MEF2C* in the iNs cell line resulted in reduced expression of *BNIP3L*. *BNIP3L* is involved in the selective removal of damaged mitochondria through a process called mitophagy. *BNIP3L* downregulation induces synaptic dysfunction arising from the accumulation of damaged mitochondria that leads to reduced mitochondrial respiration function and synaptic density [77]. It has been reported that mitophagy is signifi-cantly impaired in neurodegenerative disorders including Alzheimer's disease, Parkinson's disease, amyotrophic lateral sclerosis and Huntington's [78–81]. A recent investigation has identified both common and rare mutations in the *BNIP3L* gene in individuals diagnosed with SCZ [82]. The effect of identified genome-wide significant SNPs at *MEF2C* on its function remains to be elucidated but here is evidence that these variants may have downstream effects on direct targets of *MEF2C*, in this case potentially dysregulating *BNIP3L* and potentially con-tributing to mitochondrial dysfunction.

A first limitation of this study is that the ChIP-seq data was not generated from the same human neural cell models as the RNA-seq data, it came from human fetal brain cultures. Ide-ally, these data would come from the same source when trying to combine them to identify direct target genes. In addition, it would have strengthened the study to have validated the ChIP-seq and RNA-seq results at some target genes with quantitative PCR. It is noteworthy that the three gene-sets that were enriched for common and rare variants associated with neu-rodevelopmental disorders and phenotypes were also the largest gene-sets (all >1,000 genes) whereas the other 5 gene-sets each contained <500 genes. Therefore, we likely had greater sta-tistical power to detect enrichments in these larger gene-sets. The smaller gene-sets were all enriched for variants associated with SCZ, IQ or EA at least nominally significant levels and thus may also index relevant functions to these phenotypes.

In conclusion, our study leverages data from human neural cell models of *MEF2C* to inves-tigate putative molecular mechanisms of SCZ and cognitive dysfunction. These include neu-ron development, metabolic processes and mitochondrial dysfunction including impaired ATP production, synaptic dysfunction, imbalance in neurotransmitter systems, and disrupted mitophagy. These mechanisms provide valuable insights into how *MEF2C* dysregulation could contribute to the development of these complex disorders. Further investigations into the

precise molecular mechanisms by which *MEF2C* and mitochondrial genes contribute to the development of these disorders are needed. Such insights may pave the way for the development of novel therapeutic strategies targeting mitochondrial pathways in the treatment of neuropsychiatric disorders.

## Supporting information

**S1 Fig. Overlap of MEF2C direct target genes in NSCs (A) and iNs (B) for different mutations and genotypes.** Venn diagrams illustrate the number of shared and unique MEF2C direct target genes in each cell type separately, across different genotypes and for different types of MEF2C gene mutation. NSCs: Neural stem cells; iNs: Induced neurons; DELhom: Homozygous deletion; DELhet: Heterozygous deletion; PB: Proximal boundary (indirect mutation of MEF2C).
(TIF)

**S1 Table. Significant DEGs in MEF2C-Disrupted NSCs and iNs.**
(XLSX)

**S2 Table. Direct Target Genes of MEF2C in NSCs and iNs Determined Using BETA Analysis.**
(XLSX)

**S3 Table. Genes Identified as Associated with SCZ (Trubetskoy et al. 2022) [7], IQ (Savage et al. 2018)[8] and EA (Lee et al. 2018) [9] by GWAS Analysis.**
(XLSX)

**S4 Table. Exome Sequencing Studies Reporting DMNs Used in this Study.**
(XLSX)

**S5 Table. Background Gene Lists Used for Gene Ontology and KEGG Pathway Enrichment Analysis.**
(XLSX)

**S6 Table. MEF2C ChIP-Seq Peak Annotation.**
(XLSX)

**S7 Table. Gene Ontology Analysis for Genes Proximal to MEF2C Peaks Identified via ChIP-seq Analysis.**
(XLSX)

**S8 Table. sLDSC Analysis Results of MEF2C Direct Target Gene-Sets Using GWAS Data for SCZ, EA, and IQ.**
(XLSX)

**S9 Table. sLDSC Analysis Results of MEF2C Direct Target Gene-Sets Using GWAS Data for SCZ, EA, and IQ, Excluding Genes Associated with IQ/EA or SCZ.**
(XLSX)

**S10 Table. sLDSC Analysis Results of MEF2C Direct Target Gene-Sets Using GWAS Data for Control Phenotypes.**
(XLSX)

**S11 Table. Gene-Set Based PRS Analysis of MEF2C Direct Target Gene-sets at Five Different PRS P-value Thresholds.**
(XLSX)

**S12 Table. Gene-Set Based PRS Analysis of MEF2C Direct Target Gene-sets at Five Different PRS P-value Thresholds, Excluding All Genes Associated with IQ or EA from the Gene-Set.**
(XLSX)

**S13 Table. Analysis of MEF2C Direct Target gene-sets Using Data on DNMs from Patients with SCZ, ASD, ID, and DD.**
(XLSX)

**S14 Table. EWCE analysis of MEF2C Direct Traget Gene-Sets in the Cameron et al. 2023 [42] scRNA-seq Data for 5 Brain Regions from Prenatal Brain.**
(XLSX)

**S15 Table. EWCE analysis of MEF2C Direct Target Gene-Sets Using Available Data From the Allen Brain Atlas (Human Multiple Cortical Areas SMART-seq) from Adult Human Brain.**
(XLSX)

**S16 Table. Gene Ontology and Pathway Enrichment Analysis for MEF2C Direct Target Gene-sets Generated from NSC_DELhet.**
(XLSX)

**S17 Table. Gene Ontology and Pathway Enrichment Analysis for MEF2C Direct Target Gene-sets Generated from NSC_DELhom.**
(XLSX)

**S18 Table. Gene Ontology and Pathway Enrichment Analysis for MEF2C Direct Target Gene-sets Generated from NSC_DELhet_PB.**
(XLSX)

**S19 Table. Gene Ontology and Pathway Enrichment Analysis for MEF2C Direct Target Gene-sets Generated from NSC_DELhom_PB.**
(XLSX)

**S20 Table. Gene Ontology and Pathway Enrichment Analysis for MEF2C Direct Target Gene-sets Generated from iNs_DELhet.**
(XLSX)

**S21 Table. Gene Ontology and Pathway Enrichment Analysis for MEF2C Direct Target Gene-sets Generated from iNs_DELhom.**
(XLSX)

**S22 Table. Gene Ontology and Pathway Enrichment Analysis for MEF2C Direct Target Gene-sets Generated from iNs_DELhet_PB.**
(XLSX)

**S23 Table. Gene Ontology and Pathway Enrichment Analysis for MEF2C Direct Target Gene-sets Generated from iNs_DELhom_PB.**
(XLSX)

**S24 Table. List of Associated Genes Found in the Enriched GO Terms.**
(XLSX)

**S25 Table. Genes Prioritized in the Latest GWAS for SCZ (Trubetskoy et al. 2022) [7].**
(XLSX)

**S26 Table. List of Genes Overlapping between GO Term Enriched Genes (S24 Table) and Genes Prioritized in the Latest SCZ GWAS (S25 Table).**
(XLSX)

**S27 Table. List of Independent Genome-Wide Significant SNPs at MEF2C SNPs Associated with SCZ, EA, and IQ.**
(XLSX)

**S28 Table. eQTL Analysis for the LD-Independent MEF2C SNPs on MEF2C Direct Target Genes Associated with Mitochondrial Function and Cell-Type Specific Expression in Different Brain Tissues Based on the GTEx Dataset.**
(XLSX)

## Acknowledgments

The authors would like to thank Profs. Bulent Ataman, Gabriella Boulting and Michael Greenberg (Harvard Medical School) for sharing *MEF2C* ChIP-seq data.

## Author Contributions

**Conceptualization:** Derek W. Morris.

**Data curation:** Deema Ali, Aodán Laighneach, Emma Corley, Saahithh Reddi Patlola, Laurena Holleran.

**Formal analysis:** Deema Ali, Aodán Laighneach, Rebecca Mahoney, Derek W. Morris.

**Funding acquisition:** Deema Ali, Declan P. McKernan, Derek W. Morris.

**Investigation:** Deema Ali, Aodán Laighneach, Emma Corley, Saahithh Reddi Patlola, Rebecca Mahoney, Laurena Holleran, John P. Kelly, Aiden P. Corvin, Brian Hallahan, Colm McDonald, Gary Donohoe, Derek W. Morris.

**Methodology:** Emma Corley, Saahithh Reddi Patlola, Rebecca Mahoney, Laurena Holleran, Derek W. Morris.

**Project administration:** Declan P. McKernan, John P. Kelly, Aiden P. Corvin, Brian Hallahan, Colm McDonald, Gary Donohoe, Derek W. Morris.

**Resources:** Declan P. McKernan, John P. Kelly, Aiden P. Corvin, Brian Hallahan, Colm McDonald, Gary Donohoe, Derek W. Morris.

**Supervision:** Declan P. McKernan, John P. Kelly, Colm McDonald, Gary Donohoe, Derek W. Morris.

**Writing – original draft:** Deema Ali, Derek W. Morris.

**Writing – review & editing:** Aodán Laighneach, Emma Corley, Saahithh Reddi Patlola, Rebecca Mahoney, Laurena Holleran, Declan P. McKernan, John P. Kelly, Aiden P. Corvin, Brian Hallahan, Colm McDonald, Gary Donohoe.

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
