## [Decision Letter · Decision Letter 0]

19 Mar 2024

Dear Dr Morris,

Thank you very much for submitting your Research Article entitled 'Direct targets of MEF2C are enriched for genes associated with schizophrenia and cognitive function and are involved in neurogenesis and mitochondrial function' to PLOS Genetics.

The manuscript was fully evaluated at the editorial level and by independent peer reviewers. The reviewers appreciated the attention to an important problem, but raised some substantial concerns about the current manuscript. Based on the reviews, we will not be able to accept this version of the manuscript, but we would be willing to review a much-revised version. We cannot, of course, promise publication at that time.

If you decide to revise the manuscript for further consideration at PLOS Genetics, please aim to resubmit within the next 60 days, unless it will take extra time to address the concerns of the reviewers, in which case we would appreciate an expected resubmission date by email to plosgenetics@plos.org.

We are sorry that we cannot be more positive about your manuscript at this stage. Please do not hesitate to contact us if you have any concerns or questions.

Yours sincerely,

Hongyan Wang, Ph.D.

Academic Editor

PLOS Genetics

Scott Williams

Section Editor

PLOS Genetics

Reviewer's Responses to Questions

**Comments to the Authors:**

Reviewer #1: This is a well-written manuscript exploring enrichment of genetic variation associated with schizophrenia and cognitive function in genomic targets of MEF2C in human neural stem cells and induced glutamatergic neurons. Variation in the MEF2C gene is itself associated with schizophrenia, cognition and developmental delay, and the finding that targets of MEF2C are also enriched for associations with these traits - and elucidation of the cells and mechanisms through which they operate - is very interesting. The authors use data from a recent study exploring transcriptional consequences of MEF2C perturbation (either direct deletion of deletion of the proximal boundary) in these cells (Mohajeri et al, AJHG 2022) and, importantly, intersect these with MEF2C ChIP-Seq data from human foetal brain to base their analyses on likely direct MEF2C targets. Their identification of a putative trans-eQTL effect of MEF2C genetic variation on prioritized schizophrenia risk gene BNIP3L is also of interest. The study appears to be carried out carefully, with a Discussion that nicely places the work in the context of existing literature.

I have the following suggestions for improvement of the manuscript:

1) The manuscript appears to be missing a dedicated Methods section. Although the methods are described to some extent in the Results (and Figure 1 provides a nice overview of the methodology employed), I think more detail of the following is necessary:

- The GWAS data used for the analyses. I’m guessing these are from refs 7-9, but these are not referenced in the relevant Results section (page 14). A dedicated methods section could also state where these data were downloaded from.

- The ChIP-Seq data.

- The SLDSR analyses (e.g. did the authors use the baseline model to derive their P-values? Are P-values 1- or 2-tailed? [the former OK as only testing enrichment].)

- The exome sequencing data for ASD, ID and DD. This should be referenced in the relevant Results section (page 18) and the Methods section could indicate where these data were downloaded from.

- Gene Ontology analyses – which tool was used for this?

- Cell type enrichment analysis - how was this carried out?

2) Page 5: ‘iNs are responsible for synthesizing glutamate….’ Suggest ‘Glutamatergic iNs synthesize glutamate…’ instead (as iNs can also be GABAergic etc).

3) Figure 1: typo ‘human feta brain tissue’

4) Page 18: ‘ASD patients’. Suggest ‘people with autism’ instead.

5) Page 25: Adult hippocampal neurogenesis in humans is still controversial, so this could be acknowledged.

6) Page 29. The authors acknowledge that use of single cell data from the adult mouse brain precluded analyses in cells from an early stage of neurodevelopment. However, there are several single cell / nuclei RNA-Seq datasets from the human prenatal brain available, which include neural progenitor cells (cycling progenitors / radial glia) similar to NSCs as well as developing glutamatergic and GABAergic neurons, deriving from regions such frontal cortex and hippocampus (e.g. PMID: 36150908), which the authors could additionally use for these analyses.

Reviewer #2: This study aims to investigate if the direct targets of MEF2C are enriched for genes containing common GWAS genetic variants associated with SCZ and cognitive function, as well as enriched for genes harboring rare de novo mutations (DNMs) contributing to neurodevelopmental disorders. Then, biological processes and specific cell types that are dysregulated due to MEF2C disruption are investigated, and finally, the authors looked into the trans-eQTLs at the MEF2C gene that are associated with altered expression of MEF2C target genes. Transcriptmics data and differentially expressed genes were previously generated by another group. Chip-seq data human fetal brain cultures were generated in this study.

Overall assessment:

the study is interesting, and the findings are relevant for describing the putative molecular mechanisms of the MEF2C gene in schizophrenia and cognitive dysfunction. However, some of the conclusions are only weakly supported by the results presented.

Below, I provide detailed comments for the authors to address, and I also suggest ways to strengthen the results and further substantiate the main conclusions.

Major points

(1) The Chip-Seq data should be analyzed and reported in greater detail. The authors should report the number of peaks, genes and functional enrichment results for the genes with an overlapping Chip-Seq signal. This will help interpreting the enrichment results that are presented later in the manuscript.

(2) It will strengthen the manuscript if the Authors validate (e.g., by Chip qPCR) some of the most relevant targets of MEF2C identified in Table 1 that are associated with genetic signals for SCZ, IQ, and EA.

(3) GWAS enrichment analysis: what is the overlap between SCZ, IQ, and EA GWAS signals? To assess specificity of the GWAS enrichments, the authors should first identify disease-specific GWAS gene sets, and similarly test the 8 DEGs for heritability enrichment.

(4) Related to the previous point, the presence a (potential) substantial overlap of GWAS signals between SCZ, IQ, and EA, might explain why the three gene-sets enriched for common variation associated with SCZ, IQ and EA also explain a significant proportion of variance in IQ. The authors should rule out this scenario before affirming that genetic variants within these gene-sets play a role in influencing both susceptibility to SCZ and cognitive performance. I suggest looking for a potential overlap of GWAS signals between SCZ, IQ, and EA, identify gene sets specific for each disease, and take these shared genes and disease-specific genes into account in their analyses.

(5) The authors should assess the stability of the results presented in Figure 3 by performing a sensitivity analysis with respect to the p-value threshold for SNP inclusion. This will strengthen the conclusions of the gene-set based PRS analysis.

(6) “We used a single-cell RNA sequencing (scRNA-seq) dataset with data from 565 cell types across nine brain regions of the mouse brain [40].” What is the rationale for using mouse data instead of human data for this analysis? Are there human single cell data that can be used for this enrichment analysis? E.g., https://celltypes.brain-map.org/ or data reported by Siletti et al., Transcriptomic diversity of cell types across the adult human brain. Science 382, eadd7046 (2023). DOI: 10.1126/science.add7046? Can cross-species differences affect the enrichments results?

(7) The background gene-set used in the functional enrichment analysis (by ClueGO) was “brain tissue-expressed genes”. Given the cell-type specific enrichments (Table 3), I suggest repeating this analysis using cell-type expressed genes. This might provide better insights into the specificity of the functional enrichments results presented in the Functional Enrichment Analysis section. I also suggest expanding this analysis beyond GO terms, and explore pathways (KEGG, Panther, Biocarta, etc) which provide information about more specific cellular processes.

(8) Furthermore, I suggest the authors to (1) present graphs with the degree of enrichment (e.g., significance, enrichment score, etc) rather than the pie charts in Supplementary Figures 2-5 and Supplementary Figures 6-9, and (2) summarise the main (key) results of the Functional Enrichment Analysis in a main figure.

(9) Trans Expression Quantitative Trait Loci Analysis. “For the five gene-sets that were enriched for association with at least one of SCZ, IQ or EA, we overlapped the MEF2C direct target genes in the enriched GO terms (Supplementary Table 15) with the 120 genes prioritized in the latest GWAS for SCZ (Supplementary Table 16) [7].” The Authors should provide a clear justification for this analysis. For example, why focusing on the latest GWAS for SCZ? Similarly, why the MEF2C direct target genes in the enriched GO terms? What is the aim of this analysis? Adding these details and explaining the rationale and hypothesis tested by this analysis will help the reader to understand the results.

(10) The identification of BNIP3L as a trans regulated gene by a SNP at MEF2C locus is of interest, but on its own this is not sufficient to support a “role of MEF2C and mitochondrial dysfunction in the etiology of the studied disorders”. Also, as pointed out by the authors, since “approximately 28% of the eQTL genes implicated in SCZ were related to mitochondria [58]”, the authors should provide more convincing evidence to strengthen the role of MEF2C in mitochondrial dysfunction. For example, I would suggest this analysis to be expanded to test the underlying hypothesis more comprehensively: e.g., are the genes from the iNs gene-sets enriched for GO terms related to mitochondrial function and energy production more likely to be regulated in trans by MEF2C? Perhaps this eQTL analysis could even be refined by accounting for genes with cell-type specific expression (enrichments presented in Table 3) to improve sensitivity.

Minor comments

(1) “SNP P values and effect sizes for SCZ were derived from a SCZ GWAS meta-analysis on 183 40,675 cases and 64,643 controls [7] (excluding Irish samples)”. What is the rationale to exclude Irish samples?

(2) “We used a single-cell RNA sequencing (scRNA-seq) dataset with data from 565 cell types across nine brain regions of the mouse brain [40].” What is the rationale for using mouse data instead of human data for this analysis? Can cross-species differences affect the enrichments results?

(3) Table 1. it will be helpful to report percentages (%) as well as absolute numbers.

Reviewer: Enrico Petretto, Duke-NUS Medical School, Singapore.

**Have all data underlying the figures and results presented in the manuscript been provided?**

Reviewer #1: Yes

Reviewer #2: Yes

PLOS authors have the option to publish the peer review history of their article (what does this mean?). If published, this will include your full peer review and any attached files.

Reviewer #1: No

Reviewer #2: **Yes: **Enrico Petretto, Duke-NUS Medical School, Singapore.

---

## [Decision Letter · Decision Letter 1]

6 Aug 2024

Dear Dr Morris,

Thank you very much for submitting your Research Article entitled 'Direct targets of MEF2C are enriched for genes associated with schizophrenia and cognitive function and are involved in neuron development and mitochondrial function' to PLOS Genetics.

The manuscript was fully evaluated at the editorial level and by independent peer reviewers. The reviewers appreciated the attention to an important topic but identified some concerns that we ask you address in a revised manuscript.

We therefore ask you to modify the manuscript according to the review recommendations. Your revisions should address the specific points made by each reviewer.

To resubmit, log into your Editorial Manager account and select the option 'Revise Submission' in the 'Submissions Needing Revision' folder.

Yours sincerely,

Hongyan Wang, Ph.D.

Academic Editor

PLOS Genetics

Scott Williams

Section Editor

PLOS Genetics

Reviewer's Responses to Questions

**Comments to the Authors:**

Reviewer #1: The authors have addressed my comments, with interesting additional results.

I have 2 suggestions for minor amendment of the text:

Lines 176 - 177: 'Enrichment for heritability was compared to the baseline model using a corresponding P value

(one-tailed to identify significant local SNP-heritability).'

I think it would be clearer to state: 'Enrichment of SNP heritability was compared to the baseline model, using the Z-score to derive a (one-tailed) P-value.'

Line 354: 'Although these sets of DEGs have been generated in differentiated neuronal cell types,...'

Aren't some DEGs taken from neural stem cells? If so, I recommend deleting this part of the sentence.

Reviewer #3: The primary aim of this research is to investigate direct target of MEFC2 which could play an important role in SCZ risk and the regulation of cognitive function. The study integrates RNA-seq data from iPSCs-derived NSCs and induced neurons, in which MEF2C function is disrupted with CHIP-seq data of MEF2C. It is a well-organized work that progressively identified that several MEF2C direct target gene-sets are enriched for SNP-based heritability for cognitive function and SCZ, as well genes containing rare de novo mutations reported in diseases studied. The specific involved biological processes and trans expression quantitative trait locus effect of a single SNP at MEF2C on the expression of a target gene, BNIP3L are also interestingly observed. This article is logical, informative and uses analyses skillfully. Suggestions for improvement of the manuscript are followed.

(3.1) Abstract: The authors state that ‘we investigated the involvement of MEF2C in these 22 phenotypes using human-derived neural stem cells (NSCs) and induced neurons (iNs)’, but in the research, only glutamatergic neurons are involved. There could be more precise formulation. Or else, incorporating other types of neurons into the study might be better.

(3.2) Figure 1: The CHIP-seq data used in this research is from human fetal brain tissue, what’s the reason for choice or, why not choose an adult one?

(3.3) More convincing evidence should be provided to make robust link of MEF2C with mitochondrial dysfunction.

(3.4) Page 21: It may be abrupt if BNIP3L is pointed out directly, I suggest the description like ‘none of the tested SNPs showed a trans eQTL effect on the expression of any gene except for the finding for BNIP3L already detected’ could be arranged forward.

**Have all data underlying the figures and results presented in the manuscript been provided?**

Reviewer #1: Yes

Reviewer #3: Yes

PLOS authors have the option to publish the peer review history of their article (what does this mean?). If published, this will include your full peer review and any attached files.

Reviewer #1: No

Reviewer #3: No

---

## [Decision Letter · Decision Letter 2]

27 Aug 2024

Dear Dr Morris,

We are pleased to inform you that your manuscript entitled "Direct targets of MEF2C are enriched for genes associated with schizophrenia and cognitive function and are involved in neuron development and mitochondrial function" has been editorially accepted for publication in PLOS Genetics. Congratulations!

Yours sincerely,

Hongyan Wang, Ph.D.

Academic Editor

PLOS Genetics

Scott Williams

Section Editor

PLOS Genetics

Comments from the reviewers (if applicable):

Reviewer's Responses to Questions

**Comments to the Authors:**

Reviewer #3: The authors have addressed all my concerned and the manuscript has been improved.

**Have all data underlying the figures and results presented in the manuscript been provided?**

Reviewer #3: Yes

PLOS authors have the option to publish the peer review history of their article (what does this mean?). If published, this will include your full peer review and any attached files.

Reviewer #3: No

**Data Deposition**

http://datadryad.org/submit?journalID=pgenetics&manu=PGENETICS-D-23-01352R2

**Press Queries**

---

## [Editor Report · Acceptance letter]

5 Sep 2024

PGENETICS-D-23-01352R2 

Direct targets of MEF2C are enriched for genes associated with schizophrenia and cognitive function and are involved in neuron development and mitochondrial function 

Dear Dr Morris, 

We are pleased to inform you that your manuscript entitled "Direct targets of MEF2C are enriched for genes associated with schizophrenia and cognitive function and are involved in neuron development and mitochondrial function" has been formally accepted for publication in PLOS Genetics! Your manuscript is now with our production department and you will be notified of the publication date in due course.

With kind regards,

Anita Estes

PLOS Genetics

On behalf of:
